# Spatially structured eco-evolutionary dynamics in a host-pathogen interaction render isolated populations vulnerable to disease

Layla Höckerstedt [1,6,7], Elina Numminen[1,7], Ben Ashby[2,3,7], Mike Boots [2,4], Anna Norberg [5] & Anna-Liisa Laine [1,5] ✉

While the negative effects that pathogens have on their hosts are well-documented in humans and agricultural systems, direct evidence of pathogen-driven impacts in wild host populations is scarce and mixed. Here, to determine how the strength of pathogen-imposed selection depends on spatial structure, we analyze growth rates across approximately 4000 host populations of a perennial plant through time coupled with data on pathogen presence-absence. We find that infection decreases growth more in the isolated than well-connected host populations. Our inoculation study reveals isolated populations to be highly susceptible to disease while connected host populations support the highest levels of resistance diversity, regardless of their disease history. A spatial eco-evolutionary model predicts that non-linearity in the costs to resistance may be critical in determining this pattern. Overall, evolutionary feedbacks define the ecological impacts of disease in spatially structured systems with host gene flow being more important than disease history in determining the outcome.

According to coevolutionary theory, hosts may evolve resistance under pathogen-imposed negative frequency-dependent selection (NFDS), whereby rare host genotypes have an advantage over the common ones[1,2]. The underlying assumptions of coevolutionary theory are the strong negative fitness effect of infection, with disease-free individuals outperforming infected ones[3], and costs of resistance that are central to maintenance of polymorphism within populations[4]. While consistent negative effects of pathogens on their host populations are well documented in humans and agricultural systems[5,6], direct evidence of pathogen-driven ecological and evolutionary change in the wild is scarce and mixed[3,7–11]. The theoretical expectation

is that the selective importance of disease is directly correlated with the frequency and severity of epidemics[12]. However, our ability to quantify the strength of pathogen-imposed selection in natural populations is limited by few available systematic spatio-temporal data on pathogen occurrence across a sufficient number of host populations.

Spatial structure and heterogeneity supported by natural host populations is in stark contrast to human-managed systems that are typically highly conductive to disease transmission due to large population sizes, high densities and low genetic variability[13]. Not surprisingly, studies focusing on wild pathosystems have revealed highly variable

[1]Organismal and Evolutionary Biology Research Program, Faculty of Biological and Environmental Sciences, 00014 University of Helsinki, Helsinki, Finland. [2]Department of Integrative Biology, University of California, Berkeley, CA 94720, USA. [3]Department of Mathematics, Simon Fraser University, Burnaby, BC V3H 5J5, Canada. [4]Biosciences, University of Exeter, Penryn TR10 9EZ, UK. [5]Department of Evolutionary Biology and Environmental Studies, University of Zürich, CH-8057, Zurich, Switzerland. [6]Present address: Finnish Meteorological Institute, FI-00101, Helsinki, Finland. [7]These authors contributed equally: Layla Höckerstedt, Elina Numminen, Ben Ashby. ✉e-mail: anna-liisa.laine@uzh.ch

disease prevalence levels. Moreover, local pathogen populations are typically ephemeral, persisting regionally as metapopulations through extinction and colonization events of local host populations[14–17]. Even when infection takes place, the fitness consequences—and the coevolutionary outcomes[18]—may vary depending on the genetic composition of the host and pathogen populations and their environment, either directly or via Genotype[HOST] × Genotype[PATHOGEN] × Environment—interactions[19,20]. Moreover, hosts in wild populations may suffer increased mortality or reduced reproduction irrespective of their infection status due to other factors such as extreme weather[21]. Hence, remarkably little is understood of how pathogens impact the fitness of their host populations in the wild.

There is increasing evidence that host-pathogen dynamics, both epidemiological and evolutionary, may be shaped by the spatial structure of the interaction[13,22–24]. Encounter rates between hosts and their pathogens are expected to be heavily influenced by connectivity to other populations, and the key metapopulation processes—gene flow, extinction, and colonization dynamics—are expected to contribute to the genetic structure of both the colonization dynamics, and the arrival of novel genetic variation into local populations[13]. As long as rates of migration are low enough to not homogenize local populations, increasing immigration is expected to increase the diversity and evolutionary potential of both host and pathogen populations[25]. While measuring migration rates in natural populations is difficult[26], population connectivity, measured as the Euclidian distances separating populations and calibrated by the species dispersal capacity, provides a powerful proxy for migration rates[27]. Consequently, spatially structured eco-evolutionary feedback dynamics may emerge, with diversity accumulating in the well-connected populations. In line with this, there is evidence of spatial structure strongly influencing how resistance is distributed, with higher resistance observed in host populations that experience higher rates of gene flow[16,28,29]. To date, it has not been established what the relative roles of gene flow vs. pathogen-imposed selection are—and how they may vary in space—in generating spatially variable patterns of resistance that have been empirically observed[16,28,29].

Here, we combine a spatial analysis of wild host-pathogen populations with an inoculation experiment, and a simulation model to understand how the ecological and evolutionary impacts of disease on host resistance vary in spatially structured populations. Specifically, we ask: (1) Is there evidence of pathogen-imposed selection on its host populations across a large, naturally fragmented host-pathogen metapopulation; (2) Does host population resistance structure, measured through an inoculation assay, reflect variable selection pressure indicated by the spatial analysis; and (3) Using a coevolutionary metapopulation model we explore how gene flow, selection, and costs of resistance contribute to the spatial structure of resistance detected with our empirical approach.

Our analysis is focused on annually recorded population size data (measured visually as coverage; m2) from some ~4000 locations of host plant *Plantago lanceolata*, and the presence-absence dynamics of its obligate fungal pathogen, *Podosphaera plantaginis*, in this host population network in the Åland islands, South-Western Finland. *Plantago lanceolata* is a perennial that produces wind-dispersed pollen, while seeds typically drop close to the mother plant. During the epidemic season, *P. plantaginis* disperses via clonally produced conidial spores that typically land within close proximity of the infected source plant[16]. The visually conspicuous symptoms caused by *P. plantaginis* enable accurate tracking of infection in the wild. Long-term epidemiological data have demonstrated this pathogen to occur as a highly dynamic metapopulation with frequent extinctions and (re) colonizations of local populations, typically persisting in any given host population only for a few years[16]. The host population spatial structure is a critical determinant of pathogen extinction-colonization dynamics: large host populations are more likely to become colonized

and to sustain inifection[16]. In contrast to predictions of the metapopulation theory[27], host population connectivity has a negative impact on pathogen colonization and persistence, suggesting these populations to vary in their suitability for the pathogen[16]. The host population network does not occur as a metapopulation[30], but is characterized by strong fluctuations in population size[31,32]. These data allow us to study whether the extent of pathogen-imposed selection depends on host population connectivity ($S^H$) and hence, evolutionary potential governed by gene flow, and whether resistance level and diversity vary among host populations depending on their degree of connectivity and disease history. Previous metapopulation models[33,34] have demonstrated the existence of overall higher resistance in well-connected populations. To better understand the mechanisms that lead to the significant interaction between population connectivity, infection history and resistance in our inoculation study, we built a host-pathogen coevolutionary metapopulation model, where we examine how different trade-off relationships impact the outcome.

Jointly our results show that the strength of pathogen-imposed selection depends on host population spatial structure. Low disease resistance in the isolated populations renders them vulnerable to pathogen attack. In the well-connected host populations high rates of gene flow associate with high resistance diversity irrespective of population disase history.

## Results

### Spatio-temporal analysis of host population growth
We used Spatial Bayesian modelling (Integrated Nested Laplace Approximation; INLA[35]) to assess how changes in host population size are influenced by the pathogen. We analyzed the relative change in host population size (m2) (defined as population size ($t$) − population size ($t$−1))/population size ($t$−1)) between consecutive years utilizing data from 2001 to 2008, i.e., eight transitions in host population size in response to pathogen presence−absence status at $t$−1. To assess whether this depends on host population connectivity, we estimated the separate effects of pathogen presence/absence in the previous year for connectivity categories—high-, low, and intermediate—that were based on the 0.2 and 0.8 quantiles of the host-connectivity values (Supplementary Fig. 1). Earlier studies have demonstrated *P. lanceolata* populations in Åland to be sensitive to drought[31,32] and hence, to reliably estimate the effect of the pathogen on host population growth rates, we included data on precipitation and field-estimated drought symptoms in our model. The model also controls for spatio-temporal autocorrelation characteristic of spatial ecological data, that may be due to unmeasured variables (e.g., habitat quality, prevailing wind-direction or other unmeasured biotic of abiotic variation), thereby providing a conservative estimate of the model parameters (Supplementary Table 1)[35].

Infection by *P. plantaginis* had a negative effect on the growth of its host populations. Across all connectivity categories, the estimated mean effects of pathogen presence on host population growth were smaller than the effects with pathogen absence, suggesting an overall negative effect of the pathogen on host-population change (Fig. 1A and Supplementary Table 1). Furthermore, the estimated mean effects of the pathogen within the connectivity categories supports the interpretation that the relative effect of the pathogen on population growth is most negative in the isolated host populations (Fig. 1A and Supplementary Table 1). The posterior uncertainty in the effects of pathogen on the population growth (indicated by the confidence intervals in Fig. 1A) are due to the nature of observational data: pathogen infections were rare at the metapopulation level in studied years, thus there is considerably more pathogen absence observations in these data (See Supplementary Table 2). The temporal autocorrelation in growth in *P. lanceolata* populations between consecutive years was estimated to be negative (Supplementary Table 1), indicating that local populations exhibit oscillatory dynamics, such that growth in one year is

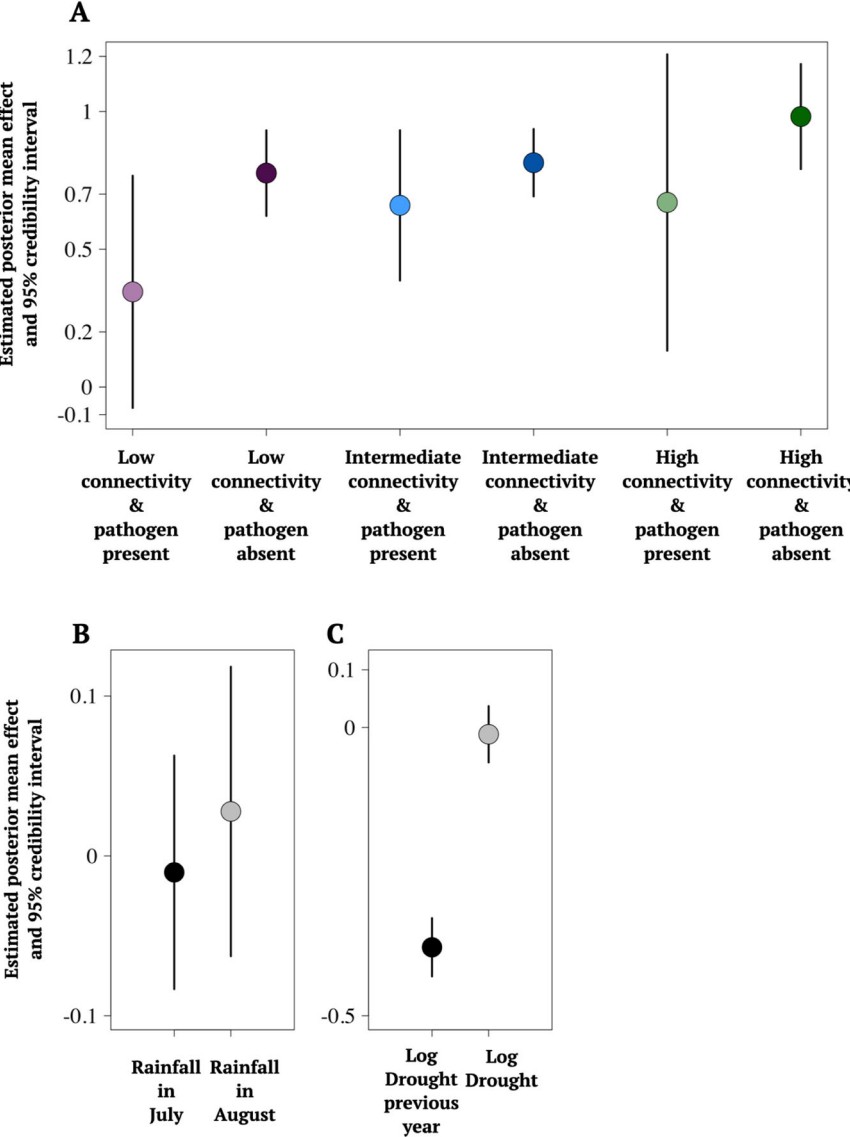

**Fig. 1 | Model estimated effects on *Plantago lanceolata* population size changes in the Åland islands in 2001–2008, based on *N* = 24042 field observations.** The estimated median effects for host population growth with 95% credibility intervals, shown with lines, of the fixed effects of the Bayesian INLA model: **A** The effect of pathogen presence and absence in the host populations in the three connectivity categories, **B** the effect of rainfall in July and August; and **C** the effect of detected drought symptoms in the host populations in the previous and current year. In the model the relative change in host population size (m2) is defined as population size ($t$ − population size ($t$−$1$))/population size ($t$−$1$)) between consecutive years utilizing data from 2001 to 2008 in response to pathogen presence-absence status at $t$ −$1$. Source data are provided as a Source Data file.

typically followed by a decline in the next year and vice versa. As many of the populations are well-established, these fluctuations could result from populations oscillating around their carrying capacities, dictated by the space and resources available for their growth. The estimated median effects for rainfall in July and August suggest that host population changes are not strongly driven by these effects, although the August rainfall had a slight positive effect on population growth (posterior mean effect 0.03, confidence interval −0.06, 0.12, Fig. 1B, Supplementary Table 1). The proportion of plants expressing drought symptoms in the previous year was significantly associated with a decline in host population size (posterior mean effect −0.38, confidence interval −0.43, −0.33, Fig. 1C, Supplementary Table 1).

**Inoculation assay quantifying host resistance phenotypes**
To examine whether the diversity and level of resistance vary among host populations depending on their degree of connectivity ($S^H$) and disease history (measured as infection status in years 2001–2014), we

performed an inoculation assay to characterize resistance phenotypes in plants sampled from 19 natural *P. lanceolata* populations. These populations occur in different locations of the host network, and were selected to represent both isolated and well-connected populations. Each plant was inoculated with four strains of *P. plantaginis* yielding resistance phenotypes values ranging between 0000 and 1111, with one depicting a resistant response and zero a susceptible response (the 16 possible resistance phenotype profiles are shown on the *x*-axis in Fig. 2A). Our inoculation study confirmed that host plants varied in their resistance against the tested powdery mildew strains (Table 1 and Fig. 2A). We were able to identify all 16 possible resistance phenotypes in the sample of 190 plants (Fig. 2A). In the connected populations, we found a greater diversity of different phenotypes, while isolated populations hosted fewer resistance phenotypes (Fig. 2A). Both the Shannon diversity index (Table 1 and Fig. 2B), and the average level of resistance (Table 1 and Fig. 2C), were higher in the well-connected than in the isolated host populations (Table 1 and Fig. 2B, C).

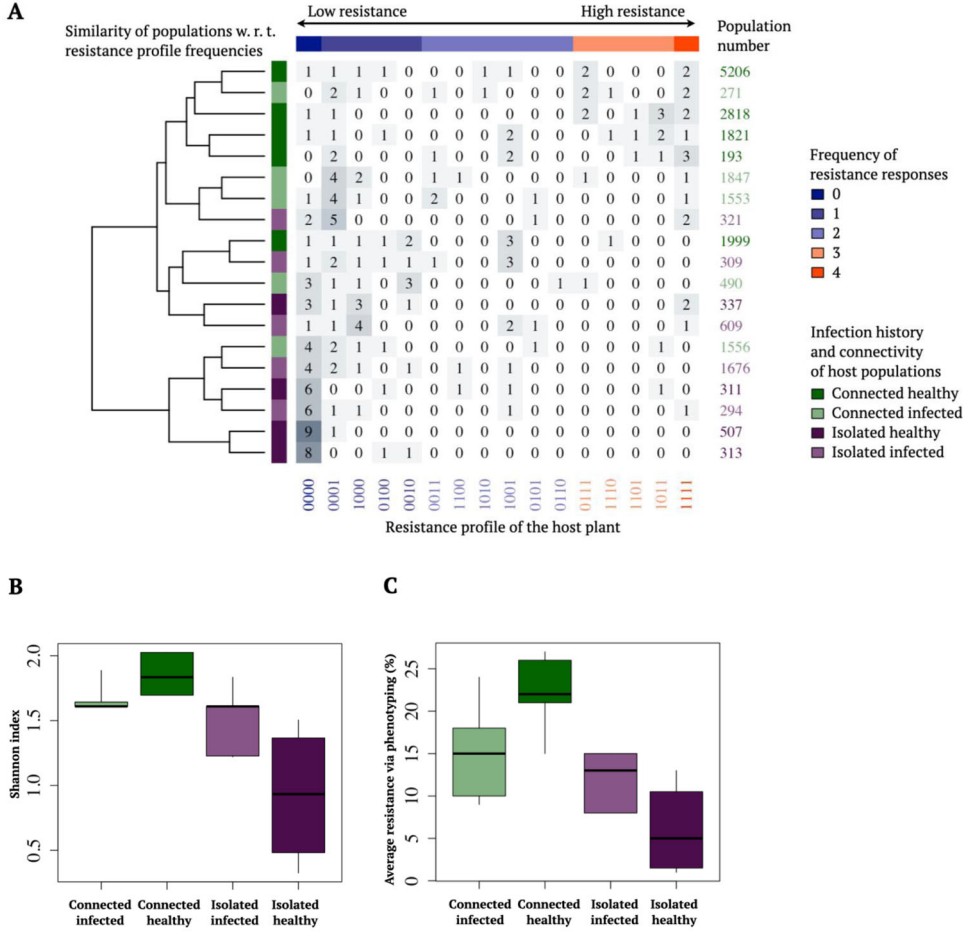

**Fig. 2 | Resistance of *Plantago lanceolata* populations depends on connectivity ($S^H$) and disease history.** The phenotype composition of the19 study populations was defined by individual plant (*n* = 10 per population) responses to the four pathogen strains, resulting in a total of *N* = 190 observations visualized in panel (**A**). The matrix of detected resistance phenotypes in the inoculation study shows clustering of similar phenotypic profiles detected in populations in each of the four connectivity ($S^H$)–infection history categories. The columns of the matrix correspond to resistance phenotypes, where the *i*'th element of the vector is 1, if resistance to pathogen strain I was detected, and zero otherwise. The rows of the matrix encode the observed frequencies of resistance phenotypes within the studied populations. The dendrogram visualizes the similarity structure between the populations, distance along the tree encoding for the degree of similarity between the populations. It is based on a hierarchical clustering (implemented with complete linkage method, aiming to find similar clusters), applied to Euclidean distances between the phenotype profiles within the populations. In panel **B** the diversity of detected 16 resistance phenotypes for the 19 study populations was characterized with a Shannon diversity index of host populations, shown separately for each connectivity ($S^H$)-disease history category, and **C** the average resistance (%) of the same populations in each category. The centre lines of the boxplots **B**, **C** show the medians, box limits show the 25 and 75% quantiles, and the whiskers span to the data extremes. Purple colours depict isolated populations, and green colours well-connected populations. Source data are provided as a Source Data file.

However, while disease history had no direct effect on phenotypic diversity nor the level of resistance, we found a significant interaction between population connectivity and infection history for both Shannon's diversity index and level of resistance (Table 1 and Fig. 2B, C). The highest diversity of phenotypes and highest resistance was measured in well-connected populations without any history of disease. In contrast, in isolated populations, we found greater diversity of resistance phenotypes and higher resistance in populations with a history of infection (Fig. 2B, C).

## The metapopulation model

We modelled both the ecological and coevolutionary dynamics of host and pathogen metapopulations by constructing the network in two stages to account for relatively well and poorly connected demes (see "Methods"). We modelled the genetics of the system using a multilocus gene-for-gene framework[36] with haploid host and pathogen genotypes characterized by *L* biallelic loci, where 0 and 1 represent the presence and absence, respectively, of resistance and infectivity alleles. Hosts and pathogen with more resistance or infectivity alleles are assumed to pay higher fitness costs, as defined in the methods. We ran 200 stochastic simulations using the tau-leap method[37] for each of the parameter sets described in Supplementary Table 3 (example simulation dynamics are shown in Fig. 3D–F). On average, disease prevalence (*D*), resistance (*R*), and infectivity (*I*) were always higher in well-connected than in poorly connected populations regardless of metapopulation structure, transmissibility of the pathogen, or the nature of the trade-offs (Supplementary Table 3). However, the difference between well and poorly connected populations was generally greater when: (1) the metapopulation structure was assortative (i.e., well connected populations are more likely to be connected to other well connected populations than by chance) than random; (2) the pathogen was more transmissible; or (3) host resistance was associated with fitness costs that diminish as resistance increases (i.e., costs of resistance decelerate, $c_H^2 < 0$) (Supplementary Table 4). Overall, we found that the pattern of the empirical results shown in Fig. 2C was most likely to

**Table 1 | The effects and effect sizes of connectivity and disease history on resistance diversity (Shannon diversity), and the average level of resistance in the 19 studied *Plantago lanceolata* populations**

| Source (Shannon diversity) | d.f. | F | P |
|---|---|---|---|
| Connectivity | 1 | 14.95 | **0.001** |
| Disease history | 1 | 1.61 | 0.2 |
| Connectivity × disease history | 1 | 7.68 | **0.01** |
| **Shannon diversity coefficients** | | **Estimate** | **sd.** |
| Intercept | | 1.85 | 0.13 |
| History (infected) | | −0.18 | 0.18 |
| Connectivity (isolated) | | −0.93 | 0.19 |
| History (Infected) * connectivity (isolated) | | 0.76 | 0.27 |
| **Source (resistance)** | **d.f.** | **X²** | **P** |
| Connectivity | 1 | 16.55 | **<0.0001** |
| Disease history | 1 | 0.01 | 0.9 |
| Connectivity × disease history | 1 | 9.91 | **0.001** |
| Mildew strain | 3 | 36.34 | **<0.0001** |
| **Random** | | **Variance** | **sd.** |
| Population | | 0.227 | 0.477 |
| Sample (population) | | 1.206 | 1.09 |
| **Resistance fixed effects** | | **Estimate** | **sd.** |
| Intercept | | 0.5 | 0.34 |
| Connectivity (isolated) | | −2.67 | 0.53 |
| History (infected) | | −0.95 | 0.44 |
| Mildew_strain2 | | −0.86 | 0.27 |
| Mildew_Strain3 | | −0.6 | 0.26 |
| Mildew_strain4 | | 0.65 | 0.25 |
| History (infected) * connectivity (isolated) | | 2.17 | 0.69 |

Significant values are highlighted in bold.
The model fit for generalized linear mixed effect model was assessed using chi-square tests on the log-likelihood values to compare different models and significant interactions. Statistics for minimum adequate models with smallest AIC values are reported.

occur when host resistance is associated with diminishing fitness costs and is more likely for transient (Fig. 3B) than long-term dynamics (Fig. 3C).

## Discussion

Here, we show that the negative effect of pathogens on their wild host populations depends on spatial structure. This finding suggests that the strength of pathogen-imposed selection may vary across space in a predictable manner. Overall, finding a consistent negative effect of infection on host population growth is noteworthy given the myriad ecological factors that may hamper our ability to quantify costs of infection in wild populations[38]. The effect of infection on host population growth was the least negative in well-connected host populations, while isolated host populations were most vulnerable to infection, suggesting that they lack resistance diversity to effectively counter pathogen attack. Indeed, results of the inoculation study confirmed that both the diversity and the average level of resistance were higher in the well-connected than in the isolated host populations. When the interaction is characterized by strain-specific resistance such as in the interaction between *P. lanceolata* and *P. plantaginis*, resistance diversity will reduce the probability of establishment by an immigrant pathogen strain, and slow down the spread of established strains due to a mismatch between the specific avirulence alleles of pathogen and resistance alleles of host[39]. In agriculture,

even slight additions of diversity to monocultures have been shown to reduce disease levels significantly[40,41].

Theory predicts that pathogens maintain resistance polymorphism in their host populations[42–44]. As described above, our spatial statistical population model demonstrated that the isolated populations went through the strongest reductions in size—most likely through increased mortality of infected individuals[31]—which could lead to selection increasing in the frequency of resistant phenotypes locally[32]. Accordingly, in the isolated populations we measured higher resistance diversity in host populations with a history of infection than in host populations that had not been infected in the past. The effect of infection on host population growth rates in the well-connected populations was much weaker, and hence, may explain why we did not detect signs of past selection in these populations. Moreover, high rates of gene flow into the well-connected populations may swamp signatures of pathogen-imposed selection. The resulting differences in resistance among host populations is in line with previous studies that have measured higher resistance levels in well-connected host populations[16,28,29]. Jointly our results reveal that this pattern is generated by eco-evolutionary feedback resulting from spatial differences in how gene flow vs. selection drive host-pathogen dynamics in the wild. In well-connected populations, gene flow appears more important than pathogen-imposed selection in maintaining resistance diversity. In support of gene flow varying according to population connectivity, we found that population growth rates were lower in the intermediate- and low-connectivity host populations than in the well-connected host populations also in the absence of the pathogen. This suggests that increasing population isolation may also have other genetic consequences, such as lower genetic diversity and higher inbreeding depression, both of which may impact population growth rates[45,46].

In theory, polymorphism in resistance within populations is maintained by costs of resistance in the absence of the pathogen, whereas under pathogen attack, the resistant hosts outperform the susceptible ones[4]. Hence, finding high levels of resistance diversity where pathogen impact has recently been negligible may appear contrary to expectations, and suggests dispersal to be critical for maintaining variation within host populations. Our metapopulation model explored scenarios under which spatial structure, disease dynamics, and life-history trade-offs could yield similar outcomes. We find that the shape of the host trade-off was the critical predictor of whether the simulations would qualitatively match the empirical results. Our results suggest that the costs of resistance are most likely to diminish as resistance increases. Diminishing costs mean that there is an initial large cost associated with resistance and therefore it is less beneficial when disease is rare. While fitness costs associated with resistance have been widely observed, determining the shape of trade-offs from empirical data is challenging, especially when trade-offs are close to linear or vary with environment, and it is impossible to determine trade-off shapes when only two host phenotypes are compared (as is often the case). However, experimental evolution of bacteria and phages has demonstrated that decelerating costs of resistance are possible[4]. In addition, our simulations suggest that the pattern detected in the empirical results is most likely to occur prior to the system reaching equilibrium and when metapopulation connectivity is assortative. The fact that the transient simulations dynamics tend to provide a better qualitative match to the empirical results does not imply that the resistance patterns detected in the archipelago will necessarily fade in the long-term (many simulations were qualitative matches at equilibrium), although our model indicates that this is a possibility. We think that it is interesting to note that the patterns we see are found for a wider range of parameter values under transient dynamics, but we get the same inference of the key characteristics that lead to the patterns we see. Whether or not the patterns are only transient is an empirical question.

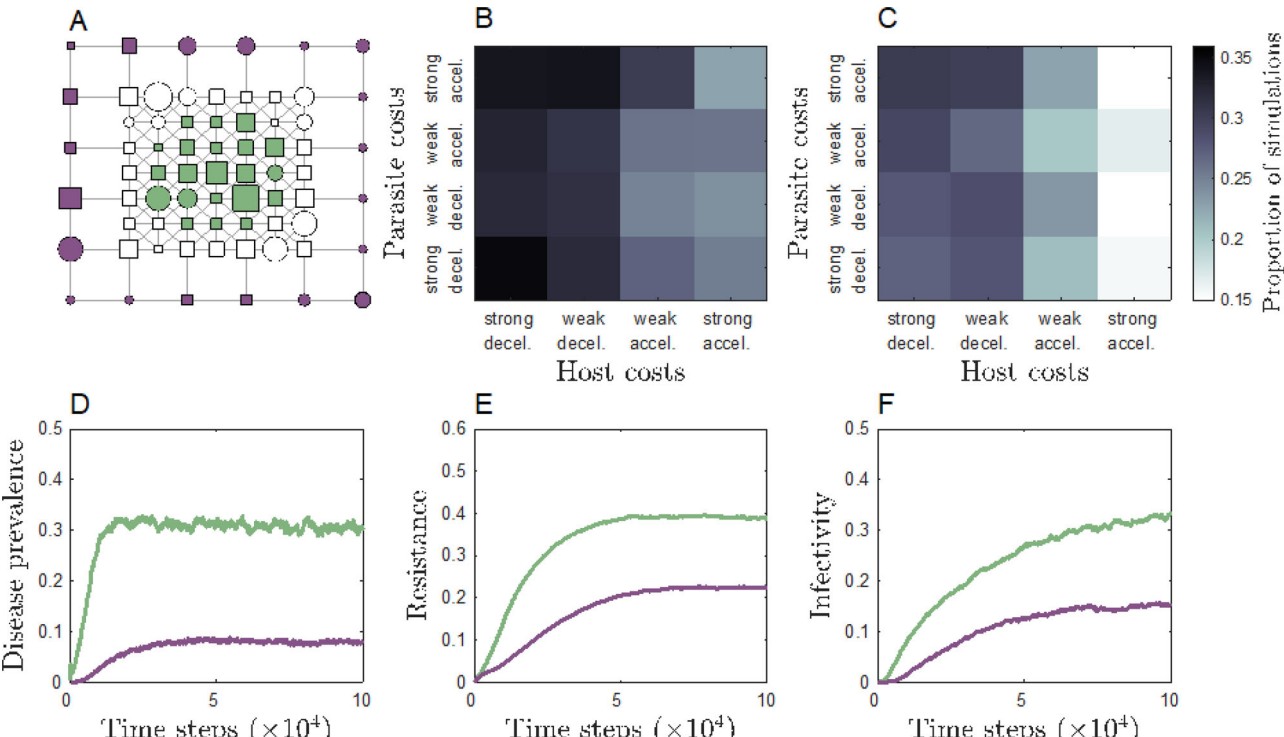

**Fig. 3 | Metapopulation simulation results. A** Example snapshot of the simulation dynamics at $t = 10{,}000$ across a metapopulation with assortative connectivity, highlighting well (green) and poorly (purple) connected populations (unshaded populations are neither well nor poorly connected) that are currently infected (squares) and uninfected (circles). The size of each node corresponds to the mean resistance of the local population. Proportion of simulations which qualitatively match the empirical results as the shape of the host and pathogen cost functions are varied for transient (**B**) and long-term (**C**) dynamics: (strong decel. (decelerating): $c_H^2$, $c_P^2 = -10$; weak decel.: $c_H^2$, $c_P^2 = -3$; weak accel. (accelerating): $c_H^2$, $c_P^2 = 3$; strong accel.: $c_H^2$, $c_P^2 = 10$). **D–F** Example simulation results, showing mean (bold line) and standard deviations (shading) for disease prevalence, i.e., proportion of infected hosts (**D**), resistance (**E**), and infectivity, i.e., the average proportion of loci with infectivity alleles in the parasite population (**F**) in well (green) and poorly (purple) connected populations ($c_H^2 = -3$, $c_P^2 = 10$, $\beta = 0.01$, with assortative network structure). Fixed parameters as defined in Supplementary Table 3.

Together, our results show how spatial fragmentation leading to the isolation of host populations drives the loss of diversity and increases host vulnerability to infectious diseases. By combining field population data with a controlled inoculation assay and a simulation model we demonstrate how spatial structure generates variation in the strength of pathogen-imposed selection, and thus provides a compelling example of how landscape fragmentation drives epidemiological and coevolutionary processes in nature.

## Methods
### The pathosystem

*Plantago lanceolata* L. is a perennial monoecious ribwort plantain that reproduces both clonally via the production side rosettes, and sexually via wind pollination. Seeds drop close to the mother plant and usually form a long-term seed bank[47]. *Podosphaera plantaginis* (Castagne; U. Braun and S. Takamatsu) (*Erysiphales*, Ascomycota) is an obligate biotrophic powdery mildew that infects only *P. lanceolata* and requires living host tissue through its life cycle[48]. It completes its life cycle as localized lesions on host leaves, only the haustorial feeding roots penetrating the leaf tissue to feed nutrients from its host. Infection causes significant stress for host plant and may increase the host mortality[31]. The interaction between *P. lanceolata* and *P. plantaginis* is strain-specific, whereby the same host genotype may be susceptible to some pathogen genotypes while being resistant to others[49]. The putative resistance mechanism includes two steps. First, resistance occurs when the host plant first recognizes the attacking pathogen and blocks its growth. When the first step fails and infection takes place, the host may mitigate infection development. Both resistance traits vary among host genotypes[49].

Approximately 4000 *P. lanceolata* populations form a network covering an area of 50 × 70 km in the Åland Islands, SW of Finland. Disease incidence (0/1) in these populations has been recorded systematically every year in early September since 2001 by approximately 40 field assistants, who record the occurrence of the fungus *P. plantaginis* in the local *P. lanceolata* populations[30]. At this time, disease symptoms are conspicuous as infected plants are covered by white mycelia and conidia. The coverage (m²) of *P. lanceolata* in the meadows was recorded between 2001 and 2008 and is used as an estimate of host population size. In the field survey two technicians estimate *Plantago* population size by visually estimating how much ground/other vegetation *P. lanceolata* foliage covers (m²) in each meadow. The proportion of *P. lanceolata* plants in each population suffering from drought is also estimated annually in the survey. Data on average rainfall (mm) in July and August was estimated separately for each population using detailed radar-measured rainfall (obtained by Finnish Meteorological Institute) and it was available for years 2001–2008.

Host population connectivity ($S^H$)[27] for each local population $i$ was computed with the formula that takes into account the area of host coverage (m²) of all host populations surveyed, denoted with ($A_j$), and their spatial location compared to other host populations. We assume that the distribution of dispersal distances from a location are described by negative exponential distribution. Under this assumption, the following formula (1) quantifies for a focal population $i$, the effect of all other host populations, taking into account their population sizes and how strongly they are connected

through immigration to it:

$$S_i^H = \sum_{j \neq i} e^{-\alpha d_{ij}} \sqrt{A_j}. \tag{1}$$

here, $d_{ij}$ is the Euclidian distance between populations $i$ and $j$ and $1/\alpha$ equals the mean dispersal distance, which was set to be two kilometres based on results from a previous study[16].

The annual survey data has demonstrated that *P. plantaginis* infects annually 2–16% of all host populations and persists as a highly dynamic metapopulation through extinctions and re-colonizations of local populations[16]. The number of host populations has remained relatively stable over the study period[49]. The first visible symptoms of *P. plantaginis* infection appear in late June as white-greyish lesions consisting of mycelium supporting the dispersal spores (conidia) that are carried by wind to the same or new host plants. Six to eight clonally produced generations follow one another in rapid succession, often leading to local epidemic with substantial proportion of the infected hosts by late summer within the host local population. *Podosphaera plantaginis* produces resting structures, chasmothecia, that appear towards the end of growing season in August–September[31]. Between 20% and 90% of the local pathogen populations go extinct during the winter, and thus the recolonization events play an important role in the persistence of the pathogen regionally[16].

### Inoculation assay: Effect of connectivity and disease history on phenotypic disease resistance

**Host and pathogen material for the experiment.** To examine whether the diversity and level of resistance vary among host populations depending on their degree of connectivity ($S^H$) and disease history, we selected 20 *P. lanceolata* populations for an inoculation assay. These populations occur in different locations in the host network, and were selected based on their connectivity values ($S^H$ of selected populations was 37–110 in isolated and 237–336 in highly connected category, Fig. 1). We did not include host populations in the intermediate connectivity category that was used in the population dynamic analyses in the inoculation assay due to logistic constraints. *Podosphaera plantaginis* is an obligate biotrophic pathogen that requires living host tissue throughout its life cycle, and obtaining sufficient inoculum for experiments is extremely time and space consuming. In both isolated and highly connected categories, half of the populations (IDs 193, 260, 311, 313, 337, 507, 1821, 1999, 2818 and 5206) were healthy during the study years 2001–2014, while half of the populations (IDs 271, 294, 309, 321, 490, 609, 1553, 1556, 1676 and 1847) were infected by *P. plantaginis* for several years during the same period. We collected *P. lanceolata* seeds from randomly selected ten individual plants around the patch area from each host population in August 2014.

To acquire inoculum for the assay, we collected the pathogen strains as infected leaves, one leaf from ten plant individuals from four additional host populations (IDs 3301, 4684, 1784, and 3108) in August 2014. None of the pathogen populations were same as the sampled host populations and hence, the strains used in the assay all represent allopatric combinations. Both host and pathogen populations selected for the study were separated by at least two kilometres. The collected leaves supporting infection were placed in Petri dishes on moist filter paper and stored at room temperature until later use.

Seeds from ten mother plants from each population were sown in 2:1 mixture of potting soil and sand, and grown in greenhouse conditions at 20 ± 2 °C (day) and 16 ± 2 °C (night) with 16:8 L:D photoperiod. Due to the low germination rate of collected seeds, population 260 (isolated and healthy population) was excluded from the study. Seedlings of ten different mother plants were randomly selected among the germinated plants for each population ($n = 190$), and grown in individual pots until the plants were eight weeks old.

The pathogen strains were purified through three cycles of single colony inoculations and maintained on live, susceptible leaves on Petri dishes in a growth chamber 20 ± 2 °C with 16:8 L:D photoperiod. Every two weeks, the strains were transferred to fresh *P. lanceolata* leaves. Purified powdery mildew strains (M1–M4), one representing each allopatric population (3301, 4684, 1784 and 3108), were used for the inoculation assay. To produce enough sporulating fungal material, repeated cycles of inoculations were performed before the assay.

**Inoculation assay quantifying host resistance phenotypes.** In order to study how the phenotypic resistance of hosts varies depending on population connectivity and infection history, we scored the resistance of 190 host genotypes, ten individuals from each study populations ($n = 19$), in an inoculation assay. Here, one detached leaf from each plant was exposed to a single pathogen strain (M1–M4) by brushing spores gently with a fine paintbrush onto the leaf. Leaves were placed on moist filter paper in Petri dishes and kept in a growth chamber at 20 ± 2 with a 16/8D photoperiod. All the inoculations were repeated on two individual Petri plates, leading to 760 host genotype–pathogen genotype combinations and a total of 1520 inoculations (19 populations * 10 plant genotypes * 4 pathogen strains * 2 replicates). We then observed and scored the pathogen infection on day 12 post inoculation, under dissecting microscope. The resulting plant phenotypic response was scored as 0 = susceptible (infection) when mycelium and conidia were observed on the leaf surface, and as 1 = resistance (no infection), when no developing lesions could be detected under a dissecting microscope. A genotype was defined resistant only if both inoculated replicates showed similar response (1), and susceptible if one or both replicates became infected (0).

### Statistical analyses

**Bayesian spatio-temporal INLA model of the changes in host population size.** To study how the pathogen infection influences on host population growth, we analyzed the relative change in host population size (m2) (defined as population size ($t$) − population size ($t$ −1))/population size ($t$−1)) between consecutive years utilizing data from 2001 to 2008 in response to pathogen presence-absence status at $t$−1 (Supplementary Table 2). To assess whether this depends on host population connectivity, we estimated the separate effects of pathogen presence/absence in the previous year for connectivity categories −high-, low, and intermediate−that were based on the 0.2 and 0.8 quantiles of the host-connectivity values (Fig. 1A and Supplementary Figs. 1, 2). This allowed us to directly assess and compare the effect of the pathogen on host population growth in the extreme categories between isolated and highly connected host populations which were represented in the sampling for the inoculation study (Fig. 2).

As covariates, we included the proportion (0–100%) of dry host plants measured each year within each local population as well as data on the amount of rainfall at the summer months (June, July, and August) obtained from the satellite images, as these were suggested be relevant for this pathosystem in an earlier analysis[16]. Observations where the change in host population size, or the host population coverage had absolute values larger than their 0.99 quantiles in the whole data, were regarded as outliers and omitted from the analysis. Before the analyses, all the continuous covariates were scaled and centred, and the categorical variables were transformed into binary variables.

The relative changes in local host population size between consecutive years was analyzed by a Bayesian spatio-temporal statistical model that simultaneously considers the effects of a set of biologically meaningful predictors. The linear predictor thus consists of two parts (2,3):

$$\beta X_t + z_t A_t \tag{2}$$

where $\beta$ represents the correlation coefficients corresponding to the effects of environmental covariates, $z_t$ corresponds to the spatio-temporal random effect, and $X_t$ and $A_t$ project these to the observation locations. For $z_t$ we assume that the observations from a location in consecutive time points $(t-1)$ and $t$ are described by 1st order autoregressive process:

$$z_t = \varphi z_{t-1} + w_t \qquad (3)$$

where $w_t$ corresponds to spatially structured zero-mean random noise, for which a Matern covariance function is assumed. Statistical inference then targets jointly the covariate effects $\beta$, the temporal autocorrelation $\varphi$, and the hyperparameters describing the spatial autocorrelation in $w_t$. From these the overall variance, as well as spatial range—a distance after which spatial autocorrelation ceases to be significant—can be inferred (Supplementary Fig. 3). For more detailed description of the structure of the statistical model and how to do efficient inference with it using R-INLA, we refer to refs. [16,50].

**Identification of resistance phenotypes.** The phenotype composition of each study population was defined by individual plant responses to the four pathogen strains, where each response could be "susceptible = 0" or "resistant = 1". For example, a phenotype "1111" refers to a plant resistant to all four pathogen strains. The diversity of distinct resistance phenotypes within populations was estimated using the Shannon diversity index as implemented in the *vegan* software package[51]. The Shannon diversity index for all four study groups was then analyzed using a linear model with class predictors population type (well-connected or isolated), infection history (healthy or infected), and their interaction.

**Analysis of population connectivity and infection history effects on host resistance.** To test whether host population resistance varied depending on connectivity ($S^H$) and infection history, we analyzed the inoculation responses (0 = susceptible, 1=resistant) of each host-pathogen combination by using a logit mixed-effect model in the *lme4* package[52]. The model included the binomial dependent variable (resistance-susceptible; 1/0), and class predictors population type (well-connected or isolated), infection history (healthy or infected), mildew strain (M1, M2, M3, and M4) and their interactions. Plant individual and population were defined as random effects, with plant genotype (sample) hierarchically nested under population. Model fit was assessed using chi-square tests on the log-likelihood values to compare different models and significant interactions, and the best model was selected based on AIC-values. $P$-values for regression coefficients were obtained by using the *car* package[53]. We ran all the analyses in R software[54].

**The metapopulation model**
We model the ecological and co-evolutionary dynamics of host and pathogen metapopulations to understand key features of the experimental system that impact on the qualitative patterns observed. The structure and parameters in our model are therefore not estimated using experimental data, but rather are chosen to cover a range of possibilities (e.g., low vs high transmission rates, variation in trade-off shapes for fitness costs). We construct the metapopulations in two stages to account for relatively well and poorly connected demes. All demes are identical in quality (i.e., no differences in intrinsic birth or death rates between demes) and only differ in their connectivity. Our metapopulation consists of an outer network of 20 demes, equally spaced around the unit square (0.2 units apart), and a 7×7 inner lattice of demes at a minimum distance of 0.2 units from the outer network (Fig. 3A), giving a total of 69 demes. Demes that are separated by a Euclidean distance of at most 0.2 are then connected to each other. This means that populations near the centre of the metapopulation are

highly connected, while those on the boundary of the metapopulation are poorly connected. This also has the effect of making connections between well and poorly connected demes assortative (i.e., well/poorly connected demes tend to be connected to well/poorly connected demes). We relax the assumption of assortativity in a second type of network by randomly reassigning connections between demes, while maintaining the same degree distribution. (i.e., the probability of two demes being connected is proportionate to their degree). While well connected demes still have more connections to other well connected demes than to poorly connected demes, they are not more likely to be connected to a well connected deme than by chance based on the degree distribution. In both types of network structure, we classify a deme as well-connected if it is in the top 20% of the degree distribution and poorly connected if it is in the bottom 20%.

We model the genetics using a multilocus gene-for-gene framework with haploid host and pathogen genotypes characterized by $L$ biallelic loci, where 0 and 1 represent the presence and absence, respectively, of resistance and infectivity alleles. Host genotype $i$ and pathogen genotype $j$ are represented by binary strings: $x_i^1 x_i^2 \dots x_i^L$ and $y_j^1 y_j^2 \dots y_j^L$. Resistance acts multiplicatively such that the probability of host $i$ being infected when challenged by pathogen $j$ is $Q_{ij} = \sigma^{d_{ij}}$, where $\sigma$ is the reduction in infectivity per effective resistance allele and $d_{ij} = \sum_{k=1}^{L} x_i^k (1 - y_j^k)$ is the number of effective resistance alleles (i.e., the number of loci where hosts have a resistance allele but pathogens do not have a corresponding infectivity allele). Hosts and pathogens with more resistance or infectivity alleles are assumed to pay higher fitness costs, $c_H(i)$ eq. (4) and $c_P(j)$ eq. (5) with:

$$c_H(i) = c_H^1 \left( \frac{1 - e^{\frac{c_H^2}{L} \sum_{k=1}^{L} x_i^k}}{1 - e^{c_H^2}} \right) \qquad (4)$$

and

$$c_P(j) = c_P^1 \left( \frac{1 - e^{\frac{c_P^2}{L} \sum_{k=1}^{L} y_j^k}}{1 - e^{c_P^2}} \right) \qquad (5)$$

where $0 < c_H^1, c_P^1 \le 1$ control the overall strength of the costs (i.e., the maximum proportional reduction in reproduction (hosts) or transmission rate (pathogens)) and $c_H^2, c_P^2 \in \mathbb{R}_{\ne 0}$ control the shape of the trade-off. When $c_H^2, c_P^2 < 0$ the costs decelerate (increasing returns) and when $c_H^2, c_P^2 > 0$ the costs accelerate the costs accelerate (decreasing returns) (Supplementary Fig. 4). This formulation, therefore, allows for a wide-range of trade-off shapes that may occur in nature.

The dynamics of the (finite) host and pathogen populations are modelled stochastically using the tau-leap method with a fixed step size of $\tau = 1$. For population $p$, the mean host birth rate at time $t$ for host $i$ (6) is

$$B_i^p(t) = \left( a(1 - c_H(i)) - q N_p(t) \right) S_i^p(t) \qquad (6)$$

where $a$ is the maximum per-capita birth rate, $q$ is the strength of density-dependent competition on births, $N_p(t) = S_i^p(t) + I_{i\circ}^p(t)$ is the local host population size, $S_i^p(t)$ and $I_{i\circ}^p(t) = \sum_{j=1}^{n} I_{ij}^p(t)$ are the local sizes of susceptible and infected individuals of genotype $i$, and $I_{ij}^p(t)$ is the local size of hosts of genotype $i$ infected by pathogen $j$. Host mutations occur at an average rate of $\mu_H$ per loci (limited to at most one mutation per time step), so that the mean number of mutations from host type $i$ to $i'$ is $\mu_H m_{ii'} B_i^p(t)$, where $m_{ii'} = 1$ if genotypes $i$ and $i'$ differ at exactly one locus, and is 0 otherwise.

The mean local mortalities for susceptible and infected individuals are $b S_i^p(t)$ and $(b + \alpha) I_{ij}^p(t)$, respectively, where $b$ is the natural mortality rate and $\alpha$ is the disease-associated mortality rate. The

average number of infected hosts that recover is $\gamma I_{ij}^p(t)$, where $\gamma$ is the recovery rate.

The mean number of new local infections of susceptible host type $i$ by pathogen $j$ eq. (7) is:

$$INF_{ij}^p(t) = \beta(1 - c_P(j))Q_{ij}S_i^p(t)Y_j^p(t) \tag{7}$$

where $\beta$ is the baseline transmission rate and $Y_j^p(t)$ is the local number of pathogen propagules following mutation and dispersal. Pathogen mutations occur in a similar manner to host mutations, with mutations from type $j$ to $j'$ occurring at rate $\mu_P m_{jj'} I_{\circ j}^p(t)$ where $\mu_P$ is the mutation rate per loci (limited to at most one mutation per timestep) and $I_{\circ j}^p(t) = \sum_{i=1}^n I_{ij}^p(t)$ is the local number of pathogen $j$. Following mutation, the local number of pathogens of type $j$ eq. (8) is:

$$W_j^p(t) = I_{\circ j}^p(t)(1 - \mu_P L) + \mu_P m_{jj'} I_{\circ j}^p(t) \tag{8}$$

Pathogen dispersal occurs following mutation at a rate of $\rho$ between connected demes, given by the adjacency matrix $G_{pr}$, with $G_p$ the total number of connections for deme $p$. The mean local number of pathogen propagules following mutation and dispersal eq. (9) is therefore:

$$Y_j^p(t) = W_j^p(t)\left(1 - \rho G_p\right) + \rho \sum_{r=1}^M G_{pr} W_j^r(t) \tag{9}$$

We focus our parameter sweep on: (i) the structure of the network (assortative or random connections); (ii) the strength $(c_H^1, c_P^1)$ and shape $(c_H^2, c_P^2)$ of the trade-offs; (iii) the transmission rate $(\beta)$; and (iv) the dispersal rate $(\rho)$, fixing the remaining parameters as described in Supplementary Table 1 (preliminary investigations suggested they had less of an impact on the qualitative outcome) and conducting 100 simulations per parameter set. For each simulation we initially seed all populations with the most susceptible host type and place the least infective pathogen type in one of the well-connected populations to minimize the risk of early extinction. We then solve the dynamics for 10,000 time steps (preliminary investigations indicated this was a sufficient period for the metapopulations to reach a quasi-equilibrium in terms of overall resistance). We calculate the average level of resistance (proportion of loci with a resistance allele) between time steps 4001 and 5000 (transient dynamics) and over the final 1000 time steps (long-term dynamics) for well and poorly connected demes, categorized according to whether the disease is present in (infected) or absent from (uninfected) the local population at a given time point and discarding simulations where the pathogen is driven globally extinct.

We compare the mean level of resistance in infected/uninfected poorly/well-connected populations across all simulations to the empirical results. We say that a simulation is a qualitative 'match' for the empirical findings if: (i) in poorly connected demes, the infected populations are on average at least 5% more resistant than uninfected populations; and (ii) in well-connected demes, the uninfected populations are on average at least 5% more resistant than infected populations. In other words, if $R_{CS}$ is the mean resistance for a population with connectivity $C$ ($C = W$ and $C = P$ for well and poorly connected demes, respectively) and infection status $S$ ($S = U$ and $S = I$ for uninfected and infected populations, respectively), then a parameter set is a qualitative 'match' for the empirical findings if $R_{WU} > 1.05R_{WI}$ and $1.05R_{PI} > 1.05R_{PU}$. If these criteria are not met, then the parameter set is a qualitative 'mismatch' for the empirical findings. The model is not intended to be a replica of an empirical metapopulation, but rather is used to reveal the key factors which lead to qualitatively similar distributions of resistance and disease incidences observed in the study of the Åland islands. Hence, the purpose of the model is to determine

which biological factors are likely to be crucial to the patterns observed herein.

## Reporting summary

Further information on research design is available in the Nature Research Reporting Summary linked to this article.

## Data availability

All data underlying this study are available at https://github.com/ComputerBlue/SpatialEcoEvoDynamics.[55] Source data are provided with this paper.

## Code availability

Code for population growth model is available at https://github.com/ComputerBlue/SpatialEcoEvoDynamics[55] and code for the simulation model is available at https://github.com/ecoevotheory/Hockerstedt_et_al_2022[56].

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

## Acknowledgements

We would like to acknowledge Krista Raveala and Niko Vilenius for their assistance during the experimental work and all students who participated in annual metapopulation surveys. This work was funded by grants from the Academy of Finland (334276), and the European Research Council (Consolidator Grant RESISTANCE 724508) and SNF (310030_192770/1) to A.-L.L., and LUOVA Doctoral Programme funding to L.H. M.B. acknowledges the Natural Environment Research Council (NE/J009784/1), NIH/R01-GM122061-03 and NSF-DEB- 2011109 for support. B.A. is supported by the Natural Environment Research Council (grant no. NE/N014979/1).

## Author contributions

A.-L.L., E.N., M.B., and L.H. conceived the ideas and designed the assay; L.H. conducted the experimental work and E.N., L.H., and A.N. analyzed the data. M.B. and B.A. developed and analyzed the simulation model. A.-L.L. and M.B. wrote the first draft of the manuscript. All the authors contributed to the writing of the manuscript and approved the final draft.

## Competing interests

The authors declare no competing interests.
