## [Peer Review File · Nature Communications]

Spatially structured eco-evolutionary dynamics in a host-pathogen interaction render isolated populations vulnerable to diseaseReviewers' Comments:

Reviewer #1:

Remarks to the Author:

I commend the authors on an impressive study integrating field epidemiology, experiments, and modeling! I have few minor comments:

Lines 113 – 116: Generally, discretizing a continuous variable just results in a loss of resolution/power. Is there justification that could be added for using these cutoffs? Alternatively, if it's an issue of not wanting to assume a linear relationship through the gradient, could a nonlinear relationship be fitted instead using this or a similar method, for example a polynomial or a nonlinear smooth in a Bayesian GAM?

Lines 121 – 123: I'm not sure I understand this wording, "the estimated mean effects of pathogen presence were smaller than the effects with pathogen absence". Mean effects of what? Is it meant to state that growth rates were slower? Effects of connectivity were lower? It seems from the figure that this is the effect representing population growth rate, but it's not clear in the text.

Lines 147-148: It might be helpful to give a little more explanation of "16 resistance phenotype", which is implicitly referring to susceptible/resistant vs. the 4 strains, correct?

Lines 166-167: 200 simulations per parameter set implies the model is stochastic, it would be helpful to explicitly state that in this section. If possible, a brief description of the stochastic components would be helpful too.

Reviewer #2:

Remarks to the Author:

In this manuscript Höckerstedt and colleagues explore if and how pathogen-imposed selection can impact wild host populations through time and space. The authors used a combination of empirical and modelling approaches to investigate if: 1) evidence of disease-driven selection in fragmented host populations, 2) how fragmentation affects resistance levels and 3) it is possible to model the impact of pathogen-imposed selection within host population in space and time.

The authors used a remarkable network of *P. lanceolata*/*P. plantaginis* infection records data to explore the above mentioned questions. The authors observed that isolated host populations growth was more affected by pathogen presence than well-connected populations. They also demonstrate with inoculation assays that connected populations depicted higher resistance phenotypes compared to isolated population. For well-connected populations, disease history did not impact the population resistance profiles while it had a strong impact in isolated populations supporting the common hypothesis that resistance polymorphism is maintained in presence of pathogens.

As claimed by the authors, this study is the first demonstration that the negative impact of pathogen infection depends on host spacial structure. The overall conclusions are well supported by the results. The methodology and analyses are sound, and well detailed in the methods and supplementary data to be reproducible. I recommend this paper for a publication upon addressing the few points highlighted bellow:

Major comments:

L120: One thing that I missed in the methods is how the host population growth rates were estimated in the field?

L135-140: I find it very interesting to include the rainfall and drought into the study of plant-pathogens dynamics. I wonder if the authors saw a difference in pathogen presence/absence depending on the years with higher or lower rainfalls? Would it be possible to check the effect of rainfall/drought on the pathogen infection? Can rainfalls affect the dispersal of the fungal spores - I am not familiar with the model species though?

L169: Maybe this is due to my lack of expertise in modelling but I had a hard time understanding how the pathogen transmission rate parameter was estimated in the model. Can the authors provide a bit more explanation to this?

Minor comments

L105-109: I find the introduction a bit long compared to the size of the Result section. It should be a bit shortened. For instance, the couple of sentences explaining the intended goals of the model, although it is worth to mention, would rather fit into the discussion or supplementary text in my opinion.

Figure 3: the quality of the axis labels seems low.

Reviewer #3:

Remarks to the Author:

Please see attached.

Review of “Spatially structured eco-evolutionary dynamics in a host-pathogen interaction render isolated populations vulnerable to disease”, by L. Hockerstedt et al.

Understanding the evolutionary dynamics of host-pathogen interactions in natural populations is an important and challenging problem. This manuscript describes an impressive field study of how an infectious disease (caused by an obligate fungal pathogen) influences growth rates in a host plant species (a plantain), as a function of how spatially connected local host populations are to other populations. The paper is potentially an important contribution to elucidating the evolutionary imprint of infectious disease in a heterogeneous natural system. The host population is found in scattered patches in the Aland islands. Some patches – out of the near 4000 locations, an impressively large field study -- are quite isolated, others are embedded in arrays of nearby patches. Such heterogeneity is widespread in natural populations. The authors provide crisp descriptions of the system and their field and lab methodologies. Theoretical studies have suggested that populations connected by a moderate (rather than very low) rate of dispersal (and hence gene flow) can sustain more genetic variation relevant to infection and tolerance, in both the host and pathogen. The authors note that measuring dispersal directly is difficult, but connectivity (the Euclidean distance separating populations) can provide a proxy for movement rates among occupied habitat patches. The basic goal of the study is to assess how connectivity among host populations is reflected in spatial patterns in host resistance to the fungal pathogen. The manuscript includes both field work, and a complementary coevolutionary model.

I think the results are interesting and intriguing, and like the mixture of empirical studies and theory development. I do think the manuscript needs a bit more work before being accepted.

1. The pathogen population is said to be a metapopulation (lines 54-55), with recurrent extinctions and recolonizations. The authors should describe this in more detail (or point the reader to other publications). I do not have a clear sense of the frequency of extinctions in the pathogen, and the degree to which this is in turn related to connectivity. Are there populations where the pathogen is always present, and others with frequent turnover? Is the host also a metapopulation, or are these populations persistent (over the relevant time-scale)?
2. Provide a short explanation of the connectivity measure on line 289. In particular, note that this expression assumes that density is constant across all habitat patches. There are many systems where density itself varies with habitat area. I presume there is evidence on density of *Plantago* across the study system.
3. *Plantago* is wind-dispersed, and so presumably is the fungal pathogen. In future work, the connectivity measures should be refined to take into account asymmetries in the direction of the wind (which in the Aland Islands is predominately from the south during the growing season). The authors should note this.
4. The authors use a Bayesian INLA model to assess effects of the pathogen on host population size changes. The figure legend for Fig. 1 should describe verbally what is plotted on the y-axis (lines 360 to 361 in main text). The main result here is that low connectivity with pathogen present leads to the greatest decline in population size (leftmost pink dot). However, it is noteworthy that the only populations which seem to be stable are those with high connectivity, without the pathogen. The intermediate four dots all seem to indicate populations that on average decline by the same amount. So

connectivity seems to influence growth rate, even for uninfected populations. I will return to this in #6, below.

5. Given that there are multiple samples from each population over time, I wonder why the authors didn't examine the time-series in more detail. For instance, if a pathogen colonizes, growth rates should decline, relative to the prior pathogen-free state (controlling for weather etc.). It seems like there is a lost opportunity here.

6. The patterns reported in Figure 2 B and C are very intriguing. I think the authors should consider the possibility of another population genetic process contributing to their observed patterns, namely inbreeding depression. One expects this to be greater in more isolated patches. This would reduce general tolerance of infection, on top of the specific reduction having to do with loss of specific resistance phenotypes.

7. I found the explanation of part of the patterns in Figure 2 to be unclear. In particular, why is resistance lower in connected infected, than in connected healthy? Is this because there was past infection in the now-healthy populations, boosting the frequency of resistant phenotypes? The average resistant of infected connected vs. isolated patches is roughly equivalent. Is this because of selection increasing resistant phenotypes there. These patterns need a more careful exposition in the Discussion.

8. The model is quite interesting, particularly the conclusion that there may be a diminishing cost of resistance with increasing resistance. The authors should describe some plausible mechanisms leading to this, in terms of the biology of the hosts. They note (line 226) that the empirical patterns are most likely if the system is not yet at equilibrium. Is there evidence for this, in the empirical system? Figure 1A seems to suggest that overall, the host metapopulation is in decline. The authors used a somewhat abstract landscape for the modeling work. Why not use the actual metapopulation spatial structure, and examine the same questions? This would help assess the robustness of conclusions, such as that about the diminishing cost of resistance.

Overall, a valuable contribution.

Lines 45-46: "the selective importance of disease is directly correlated with the frequency and severity of epidemics."

Responses to Referees

We want to thank the Reviewers for the constructive and helpful feedback. We have thoroughly addressed all comments, listed below point-by-point followed by our response. In the manuscript we highlight with blue font the resulting changes. We feel that these revisions have resulted in a stronger presentation of our work. Specifically, we provide more background information on the study system, and clarify methods, as well as expand our discussion to account for the valuable feedback we received.

We have updated the manuscript with a link to Github where data and code are available.

On behalf of all authors,
Anna-Liisa Laine

REVIEWER COMMENTS

Reviewer #1 (Remarks to the Author):

I commend the authors on an impressive study integrating field epidemiology, experiments, and modeling!

Thank you very much for this positive assessment of our study.

I have few minor comments:

Lines 113 – 116: Generally, discretizing a continuous variable just results in a loss of resolution/power. Is there justification that could be added for using these cutoffs? Alternatively, if it's an issue of not wanting to assume a linear relationship through the gradient, could a nonlinear relationship be fitted instead using this or a similar method, for example a polynomial or a nonlinear smooth in a Bayesian GAM?

Defining connectivity as a categorical value with the bins matching the distribution of the continuous values allowed us to analyze specifically the pathogen effect on host population growth in the extreme categories, *i.e.* isolated and highly connected populations that represent the 0.2 and 0.8 quantiles of the frequency distribution. This also allowed generating direct links between the experimental and simulation components of our study as visualized in Figure 2, as we were able to link the INLA model results directly to the sampling scheme of the inoculation assay (it would not have been possible to sample replicate populations with identical connectivity values so in our sampling scheme we resorted to high-low categories).

We have revised the text to provide motivation for this on lines 381-387: *“To assess whether this depends on host population connectivity, we estimated the separate effects of pathogen presence/absence in the previous year for connectivity categories - high-, low and intermediate – that were based on the 0.2 and 0.8 quantiles of the host-connectivity values (Fig.*

1A, Supplementary Figs.1-2). This allowed us to directly assess and compare the effect of the pathogen on host population growth in the extreme categories between isolated and highly connected host populations which were represented in the sampling for the inoculation study (Fig. 2)."

Lines 121 – 123: I'm not sure I understand this wording, "the estimated mean effects of pathogen presence were smaller than the effects with pathogen absence". Mean effects of what? Is it meant to state that growth rates were slower? Effects of connectivity were lower? It seems from the figure that this is the effect representing population growth rate, but it's not clear in the text.

Thank for noticing the unclear wording. Exactly, we mean the effect of pathogen presence on host population growth, and we now specify this on lines 128-131: "*Across all connectivity categories the estimated mean effects of pathogen presence on host population growth were smaller than the effects with pathogen absence, suggesting an overall negative effect of the pathogen on host-population change (Fig. 1A, Supplementary Table 1).*"

Lines 147-148: It might be helpful to give a little more explanation of "16 resistance phenotype", which is implicitly referring to susceptible/resistant vs. the 4 strains, correct?

Yes, exactly. We now provide more details on the phenotype characterization on lines 151-157: "*we performed an inoculation assay to characterize resistance phenotypes in plants sampled from 19 natural *P. lanceolata* populations. These populations occur in different locations of the host network, and were selected to represent both isolated and well-connected populations. Each plant was inoculated with four strains of *P. plantaginis* yielding resistance phenotype values ranging between 0000-1111, with one depicting a resistant response and zero a susceptible response (the 16 possible resistance phenotype profiles are shown on the x-axis in Figure 2A).*"

Lines 166-167: 200 simulations per parameter set implies the model is stochastic, it would be helpful to explicitly state that in this section. If possible, a brief description of the stochastic components would be helpful too.

A full description of the stochastic simulations was provided in the Methods (lines 435-514). However, we agree that adding some more details to the main text would help to clarify the nature of the model, so we have amended this sentence on lines 177-178 to: "*We ran 200 stochastic simulations using the tau-leap method³⁷ for each of the parameter sets described in Supplementary Table 3.*"

Reviewer #2 (Remarks to the Author):

In this manuscript Höckerstedt and colleagues explore if and how pathogen-imposed selection can impact wild host populations through time and space. The authors used a combination of empirical and modelling approaches to investigate if: 1) evidence of disease-driven selection in

fragmented host populations, 2) how fragmentation affects resistance levels and 3) it is possible to model the impact of pathogen-imposed selection within host population in space and time.

The authors used a remarkable network of *P. lanceolata*/*P. plantaginis* infection records data to explore the above mentioned questions. The authors observed that isolated host populations growth was more affected by pathogen presence than well-connected populations. They also demonstrate with inoculation assays that connected populations depicted higher resistance phenotypes compared to isolated population. For well-connected populations, disease history did not impact the population resistance profiles while it had a strong impact in isolated populations supporting the common hypothesis that resistance polymorphism is maintained in presence of pathogens.

As claimed by the authors, this study is the first demonstration that the negative impact of pathogen infection depends on host spacial structure. The overall conclusions are well supported by the results. The methodology and analyses are sound, and well detailed in the methods and supplementary data to be reproducible. I recommend this paper for a publication upon addressing the few points highlighted below:

Thank you very much for this positive assessment of our study and for the constructive feedback.

Major comments:

L120: One thing that I missed in the methods is how the host population growth rates were estimated in the field?

In the field, *P.lanceolata* population size is visually estimated as coverage (m²) by two field technicians in each population, and population growth rate is calculated from these data. We now clarify on lines 87-88 “*Our analysis is focused on annually recorded population size data (measured visually as coverage; m²) from some ~ 4000 locations of host plant Plantago lanceolata*” and in the methods on line 294 we now provide more details: “*In the field survey two technicians estimate Plantago population size by visually estimating how much ground/other vegetation P. lanceolata foliage covers (m²) in each meadow.*”

L135-140: I find it very interesting to include the rainfall and drought into the study of plant-pathogens dynamics. I wonder if the authors saw a difference in pathogen presence/absence depending on the years with higher or lower rainfalls? Would it be possible to check the effect of rainfall/drought on the pathogen infection? Can rainfalls affect the dispersal of the fungal spores - I am not familiar with the model species though?

Based on previous work, we expect rainfall to affect the *Plantago-Podosphaera* interaction through two different mechanisms. First, these meadows usually grow on shallow soils and plants suffer visible symptoms of drought and increased mortality during periods of low rainfall in the growing season (Laine, 2004, 2006; Salgado et al., 2020). This decline in the availability of the host is expected to have direct consequences for this obligate fungal pathogen that requires living host tissue. Hence, sufficient rain is required to maintain viable host and pathogen populations. On the other hand, heavy rains can wash away spores from the air and leaf surfaces, and indeed

rainfall in August, a critical transmission period for the pathogen, has a negative impact on disease dynamics (Jousimo et al., 2014).

Here, our motivation for including rainfall and drought data to the model was that we want to control for potential effects of abiotic conditions in host populations growth so that we could estimate the pathogen effect as reliably as possible. We now clarify this on lines 121-123: *“Earlier studies have demonstrated P. lanceolata populations in Åland to be sensitive to drought^{31,32} and hence, to reliably estimate the effect of the pathogen on host population growth rates, we included data on precipitation and field-estimated drought symptoms in our model.”*

L169: Maybe this is due to my lack of expertise in modelling but I had a hard time understanding how the pathogen transmission rate parameter was estimated in the model. Can the authors provide a bit more explanation to this?

The purpose of the model is to capture some of the key features of the real host-pathogen system (e.g. contrasting connectivity across the metapopulation, multilocus infection genetics, fitness costs of resistance and infectivity), to determine how these impact on coevolution, rather than directly model the precise disease dynamics (e.g. for making projections). The parameters used in our model are therefore not estimated using experimental data, but rather are chosen to cover a range of possibilities (e.g. low vs high transmission rates, variation in trade-off shapes for fitness costs). We have added the following text to lines 435-439 to clarify this: *“We also model the ecological and co-evolutionary dynamics of host and pathogen metapopulations to understand key features of the experimental system that impact on the qualitative patterns observed. The structure and parameters in our model are therefore not estimated using experimental data, but rather are chosen to cover a range of possibilities (e.g. low vs high transmission rates, variation in trade-off shapes for fitness costs).”*

Minor comments

L105-109: I find the introduction a bit long compared to the size of the Result section. It should be a bit shortened. For instance, the couple of sentences explaining the intended goals of the model, although it is worth to mention, would rather fit into the discussion or supplementary text in my opinion.

Thank you for this suggestion. We agree that this is not necessary information in the Introduction, and have moved this section to the Methods where we describe the simulation model, lines 524-528.

Figure 3: the quality of the axis labels seems low.

The loss of resolution appears to be due to the transfer to the figure from PDF to bitmap in Word. This should not be an issue in the PDF version.

Reviewer #3 (Remarks to the Author):

Please see attached.

Review of “Spatially structured eco-evolutionary dynamics in a host-pathogen interaction render isolated populations vulnerable to disease”, by L. Hockerstedt et al. Understanding the evolutionary dynamics of host-pathogen interactions in natural populations is an important and challenging problem. This manuscript describes an impressive field study of how an infectious disease (caused by an obligate fungal pathogen) influences growth rates in a host plant species (a plantain), as a function of how spatially connected local host populations are to other populations. The paper is potentially an important contribution to elucidating the evolutionary imprint of infectious disease in a heterogeneous natural system. The host population is found in scattered patches in the Aland islands. Some patches – out of the near 4000 locations, an impressively large field study -- are quite isolated, others are embedded in arrays of nearby patches. Such heterogeneity is widespread in natural populations. The authors provide crisp descriptions of the system and their field and lab methodologies. Theoretical studies have suggested that populations connected by a moderate (rather than very low) rate of dispersal (and hence gene flow) can sustain more genetic variation relevant to infection and tolerance, in both the host and pathogen. The authors note that measuring dispersal directly is difficult, but connectivity (the Euclidean distance separating populations) can provide a proxy for movement rates among occupied habitat patches. The basic goal of the study is to assess how connectivity among host populations is reflected in spatial patterns in host resistance to the fungal pathogen. The manuscript includes both field work, and a complementary coevolutionary model.

I think the results are interesting and intriguing, and like the mixture of empirical studies and theory development. I do think the manuscript needs a bit more work before being accepted.

Thank you for the positive assessment of our study, and for the constructive feedback to help us improve our work.

1. The pathogen population is said to be a metapopulation (lines 54-55), with recurrent extinctions and recolonizations. The authors should describe this in more detail (or point the reader to other publications). I do not have a clear sense of the frequency of extinctions in the pathogen, and the degree to which this is in turn related to connectivity. Are there populations where the pathogen is always present, and others with frequent turnover? Is the host also a metapopulation, or are these populations persistent (over the relevant time-scale)?

We agree that these dynamics are important for the current study, and we now expand on our description on lines 94-102: “*Long-term epidemiological data have demonstrated this pathogen to occur as a highly dynamic metapopulation with frequent extinctions and (re)colonizations of local populations, typically persisting in any given host population only for a few years¹⁶ The host population spatial structure is a critical determinant of pathogen extinction-colonization dynamics: large host populations are more likely to become colonized and sustain infection¹⁶. In contrast to predictions of the metapopulation theory²⁷, host population connectivity has a negative impact on pathogen colonization and persistence, suggesting these populations to vary in their suitability for the pathogen¹⁶. The host population network does not occur as a metapopulation³⁰, but is characterized by strong fluctuations in population size^{31,32}.*”

The most comprehensive analysis of the pathogen population dynamics to date is: Jousimo, J, Tack, AJM, Ovaskainen, O., Mononen, T., Susi, H., Tollenaere, C. & Laine, A.-L.

(2014) Ecological and evolutionary effects of fragmentation on infectious disease dynamics. *Science*, 344: 1289-1293. doi:10.1126/science.1253621, which is citation #16 in our manuscript.

2. Provide a short explanation of the connectivity measure on line 289. In particular, note that this expression assumes that density is constant across all habitat patches. There are many systems where density itself varies with habitat area. I presume there is evidence on density of *Plantago* across the study system.

We provide a description the connectivity measure on lines 301-311: “*Host population connectivity (S^H)²⁷ for each local population i was computed with the formula that takes into account the area of host coverage (m^2) of all host populations surveyed, denoted with (A_j), and their spatial location compared to other host populations. We assume that the distribution of dispersal distances from a location are described by negative exponential distribution. Under this assumption, the following formula quantifies for a focal population i , the effect of all other host populations, taking into account their population sizes and how strongly they are connected through immigration to it:*

$$S_i^H = \sum_{j \neq i} e^{-\alpha d_{ij}} \sqrt{A_j}.$$

Here, d_{ij} is the Euclidian distance between populations i and j and $1/\alpha$ equals the mean dispersal distance, which was set to be two kilometers based on results from a previous study¹⁶”.

It would be possible to calculate host population density using data *Plantago* coverage (m^2) and habitat area. However, this would not capture the sometimes aggregated manner in which *Plantago* occurs within populations. Here, we use the connectivity measurement as a proxy for incoming gene flow into the focal populations, and for this we expect population size to be more important than density.

3. *Plantago* is wind-dispersed, and so presumably is the fungal pathogen. In future work, the connectivity measures should be refined to take into account asymmetries in the direction of the wind (which in the Aland Islands is predominately from the south during the growing season). The authors should note this.

The pollen of *Plantago* is wind-dispersed, while the seed simply drop to the ground (Lines 275-276): “*Plantago lanceolata* L. is a perennial monoecious ribwort plantain that reproduces both clonally via the production side rosettes, and sexually via wind pollination”.

The fungal pathogen is also wind-dispersed, and we have added this relevant information that was missing from the manuscript to lines 315-317: “*The first visible symptoms of P. plantaginis* infection appear in late June as white-greyish lesions consisting of mycelium supporting the dispersal spores (conidia) that are carried by wind to the same or new host plants.”

An earlier analysis found wind direction to have an impact on the pathogen dynamics (Laine & Hanski, 2006), but this signal was not detected in a follow-up study (Jousimo et al. 2014). It would indeed be interesting to test different connectivity measures that account for this. In our analyses, we control for spatio-temporal autocorrelation that may be due to unmeasured variables, thereby providing a conservative estimate of the model parameters. We now specify in

the text on lines 123-127: *“The model also controls for spatio-temporal autocorrelation characteristic of spatial ecological data, that may be due to unmeasured variables (e.g. habitat quality, prevailing wind-direction or other unmeasured biotic or abiotic variation), thereby providing a conservative estimate of the model parameters (Supplementary Table 1) ³⁵. ”*

4. The authors use a Bayesian INLA model to assess effects of the pathogen on host population size changes. The figure legend for Fig. 1 should describe verbally what is plotted on the y-axis (lines 360 to 361 in main text). The main result here is that low connectivity with pathogen present leads to the greatest decline in population size (leftmost pink dot). However, it is noteworthy that the only populations which seem to be stable are those with high connectivity, without the pathogen. The intermediate four dots all seem to indicate populations that on average decline by the same amount. So connectivity seems to influence growth rate, even for uninfected populations. I will return to this in #6, below.

Thank you for this helpful suggestion. We have updated the figure legend to specify: *“In the model the relative change in host population size (m_2) is defined as population size (t) - population size ($t-1$) / population size ($t-1$) between consecutive years utilizing data from 2001-2008 in response to pathogen presence-absence status at $t-1$. ”*

This is an important observation, and we have added discussion on this point to 222-226: *“In support of gene flow varying according to population connectivity, we found that population growth rates were lower in the intermediate- and low-connectivity host populations than in the well-connected host populations also in the absence of the pathogen. This suggests that increasing population isolation may also have other genetic consequences, such as lower genetic diversity and higher inbreeding depression, both of which may impact population growth rates ^{45,46}. ”*

5. Given that there are multiple samples from each population over time, I wonder why the authors didn't examine the time-series in more detail. For instance, if a pathogen colonizes, growth rates should decline, relative to the prior pathogen-free state (controlling for weather etc.). It seems like there is a lost opportunity here.

Thank you for pointing out this potential source of confusion. This is exactly what our model does, and we have revised the text to clarify this on lines 113-117: *“We used Spatial Bayesian modelling (Integrated Nested Laplace Approximation; INLA³⁵) to assess how changes in host population size are influenced by the pathogen. We analyzed the relative change in host population size (m_2) (defined as population size (t) - population size ($t-1$) / population size ($t-1$)) between consecutive years utilizing data from 2001-2008, i.e. eight transitions in host population size in response to pathogen presence-absence status at $t-1$. ”*

6. The patterns reported in Figure 2 B and C are very intriguing. I think the authors should consider the possibility of another population genetic process contributing to their observed patterns, namely inbreeding depression. One expects this to be greater in more isolated patches. This would reduce general tolerance of infection, on top of the specific reduction having to do with loss of specific resistance phenotypes.

Thank you for suggesting this very interesting possibility that we had not discussed. *Plantago lanceolata* is obligately outcrossing and we do not have data on potential inbreeding, but we fully agree that isolated populations may have lower genetic diversity and higher inbreeding depression, both of which could contribute to population growth rates. We now discuss this possibility on lines 222-226: “*In support of gene flow varying according to population connectivity, we found that population growth rates were lower in the intermediate- and low-connectivity host populations than in the well-connected host populations also in the absence of the pathogen. This suggests that increasing population isolation may also have other genetic consequences, such as lower genetic diversity and higher inbreeding depression, both of which may impact population growth rates* ^{45,46}.”

7. I found the explanation of part of the patterns in Figure 2 to be unclear. In particular, why is resistance lower in connected infected, than in connected healthy? Is this because there was past infection in the now-healthy populations, boosting the frequency of resistant phenotypes? The average resistant of infected connected vs. isolated patches is roughly equivalent. Is this because of selection increasing resistant phenotypes there. These patterns need a more careful exposition in the Discussion.

We believe that this pattern jointly reflects ecological and evolutionary dynamics: In the well-connected populations gene flow maintains high diversity in resistance. The uninfected well-connected populations are likely to be so diverse that it is difficult for the pathogen to establish in these populations (*cf.* Jousimo *et al* 2014: Lower pathogen colonization rates in well-connected host populations). In the infected well-connected populations although the pathogen has successfully established, our INLA model reveals that it has a limited effect on the host population, i.e. selection for resistance is not very high (most likely due to the fact that host diversity limits pathogen population size). Furthermore, the effects of selection may be swamped by high rates of gene flow. In the isolated populations the INLA model reveals the pathogen to have a strong effect on host population growth, i.e. the strength of pathogen-imposed selection is strong, and the signatures of pathogen imposed selection are not swamped by gene flow.

We have revised the section on lines 207-226 to better describe this: “*Theory predicts that pathogens maintain resistance polymorphism in their host populations* ⁴²⁻⁴⁴. *As described above, our spatial statistical population model demonstrated that the isolated populations went through the strongest reductions in size - most likely through increased mortality of infected individuals*³¹ - *which could lead to selection increasing in the frequency of resistant phenotypes locally*³². *Accordingly, in the isolated populations we measured higher resistance diversity in host populations with a history of infection than in host populations that had not been infected in the past. The effect of infection on host population growth rates in the well-connected populations was much weaker, and hence, may explain why we did not detect signs of past selection in these populations. Moreover, high rates of gene flow into the well-connected populations may swamp signature of pathogen-imposed selection. The resulting differences in resistance among host populations is in line with previous studies that have measured higher resistance levels in well-connected host populations* ^{16,28,29}. *Jointly our results reveal that this pattern is generated by eco-evolutionary feedback resulting from spatial differences in how gene flow vs. selection drive host-pathogen dynamics in the in the wild. In the well-connected populations gene flow appears more important than pathogen-imposed selection in maintaining resistance diversity. In support of*

gene flow varying according to population connectivity, we found that population growth rates were lower in the intermediate- and low-connectivity host populations than in the well-connected host populations also in the absence of the pathogen. This suggests that increasing population isolation may also have other genetic consequences, such as lower genetic diversity and higher inbreeding depression, both of which may impact population growth rates^{45,46}.”

8. The model is quite interesting, particularly the conclusion that there may be a diminishing cost of resistance with increasing resistance. The authors should describe some plausible mechanisms leading to this, in terms of the biology of the hosts. They note (line 226) that the empirical patterns are most likely if the system is not yet at equilibrium. Is there evidence for this, in the empirical system? Figure 1A seems to suggest that overall, the host metapopulation is in decline. The authors used a somewhat abstract landscape for the modeling work. Why not use the actual metapopulation spatial structure, and examine the same questions? This would help assess the robustness of conclusions, such as that about the diminishing cost of resistance.

The purpose of the model is to capture some of the key features of the real host-pathogen system (e.g. contrasting connectivity across the metapopulation, multilocus infection genetics, fitness costs of resistance and infectivity), to determine how these impact on coevolution, rather than directly model the precise dynamics of this system. The advantage of this approach is that it allows us to generalize the empirical patterns beyond this particular system to determine the key features of host-pathogen metapopulations that create qualitatively similar outcomes.

We therefore use a simplified/abstract metapopulation structure rather than attempt to replicate the actual metapopulation structure, because it allows us to identify which features are likely to lead to qualitatively similar outcomes in other systems. If we were to use the actual metapopulation structure, then we would not know whether there is something special about the actual metapopulation structure that produces these patterns. The fact that we can recreate the same qualitative dynamics in a simplified metapopulation tells us that the patterns observed do not depend on the actual metapopulation structure and so are likely not unique to this system. The empirical system is highly dynamic, and is characterized by strong fluctuations in pathogen occurrence patterns over time, as well as a pronounced change in spatial synchrony over the study period (Jousimo et al., 2014; Penczykowski et al., 2015).

We have added the following text to lines 435-439 to clarify the purpose of our model: *“We model the ecological and co-evolutionary dynamics of host and pathogen metapopulations to understand key features of the experimental system that impact on the qualitative patterns observed. The structure and parameters in our model are therefore not estimated using experimental data, but rather are chosen to cover a range of possibilities (e.g. low vs high transmission rates, variation in trade-off shapes for fitness costs).”*

Overall, a valuable contribution.

Thank you again for the valuable and constructive feedback on our study.

Lines 45-46: “the selective importance of disease is directly correlated with the frequency and severity of epidemics.”

Thank you for noticing, we have corrected this.

References

- Jousimo, J., Tack, A. J. M., Ovaskainen, O., Mononen, T., Susi, H., Tollenaere, C., & Laine, A.-L. (2014). Disease ecology. Ecological and evolutionary effects of fragmentation on infectious disease dynamics. *Science (New York, N.Y.)*, *344*(6189), 1289–1293. <https://doi.org/10.1126/science.1253621>
- Laine, A.-L. (2004). Resistance variation within and among host populations in a plant–pathogen metapopulation: Implications for regional pathogen dynamics. *Journal of Ecology*, *92*(6), 990–1000. <https://doi.org/10.1111/j.0022-0477.2004.00925.x>
- Laine, A.-L. (2006). Evolution of host resistance: Looking for coevolutionary hotspots at small spatial scales. *Proceedings of the Royal Society B: Biological Sciences*, *273*(1584), 267–273. <https://doi.org/10.1098/rspb.2005.3303>
- Laine, A.-L., & Hanski, I. (2006). Large-Scale Spatial Dynamics of a Specialist plant Pathogen in a Fragmented Landscape. *Journal of Ecology*, *94*(1), 217–226.
- Penczykowski, R. M., Walker, E., Soubeyrand, S., & Laine, A.-L. (2015). Linking winter conditions to regional disease dynamics in a wild plant–pathogen metapopulation. *New Phytologist*, *205*(3), 1142–1152. <https://doi.org/10.1111/nph.13145>
- Salgado, A. L., DiLeo, M. F., & Saastamoinen, M. (2020). Narrow oviposition preference of an insect herbivore risks survival under conditions of severe drought. *Functional Ecology*, *34*(7), 1358–1369. <https://doi.org/10.1111/1365-2435.13587>

Reviewers' Comments:

Reviewer #1:

Remarks to the Author:

Thank you for clearly addressing my prior comments. The manuscript will be an exciting contribution to the field!

Reviewer #2:

Remarks to the Author:

All my comments have been addressed by the authors in this manuscript revisions.